# First Results of a New Deep Learning Reconstruction Algorithm on Image Quality and Liver Metastasis Conspicuity for Abdominal Low-Dose CT

**DOI:** 10.3390/diagnostics13061182

**Published:** 2023-03-20

**Authors:** Joël Greffier, Quentin Durand, Chris Serrand, Renaud Sales, Fabien de Oliveira, Jean-Paul Beregi, Djamel Dabli, Julien Frandon

**Affiliations:** 1IMAGINE UR UM 103, Department of Medical Imaging, Nimes University Hospital, Montpellier University, 30029 Nimes, France; 2Department of Biostatistics, Clinical Epidemiology, Public Health, and Innovation in Methodology (BESPIM), CHU Nimes, 30029 Nimes, France

**Keywords:** artificial intelligence, deep learning, multidetector computed tomography, image enhancement, liver neoplasms

## Abstract

The study’s aim was to assess the impact of a deep learning image reconstruction algorithm (Precise Image; DLR) on image quality and liver metastasis conspicuity compared with an iterative reconstruction algorithm (IR). This retrospective study included all consecutive patients with at least one liver metastasis having been diagnosed between December 2021 and February 2022. Images were reconstructed using level 4 of the IR algorithm (i4) and the Standard/Smooth/Smoother levels of the DLR algorithm. Mean attenuation and standard deviation were measured by placing the ROIs in the fat, muscle, healthy liver, and liver tumor. Two radiologists assessed the image noise and image smoothing, overall image quality, and lesion conspicuity using Likert scales. The study included 30 patients (mean age 70.4 ± 9.8 years, 17 men). The mean CTDI_vol_ was 6.3 ± 2.1 mGy, and the mean dose-length product 314.7 ± 105.7 mGy.cm. Compared with i4, the HU values were similar in the DLR algorithm at all levels for all tissues studied. For each tissue, the image noise significantly decreased with DLR compared with i4 (*p* < 0.01) and significantly decreased from Standard to Smooth (−26 ± 10%; *p* < 0.01) and from Smooth to Smoother (−37 ± 8%; *p* < 0.01). The subjective image assessment confirmed that the image noise significantly decreased between i4 and DLR (*p* < 0.01) and from the Standard to Smoother levels (*p* < 0.01), but the opposite occurred for the image smoothing. The highest scores for overall image quality and conspicuity were found for the Smooth and Smoother levels.

## 1. Introduction

Patients with liver metastasis undergo repeated CT scans during their follow-ups, which have cumulative doses that sometimes exceed 100 mSv [1]. Dose optimization is therefore an important challenge. Dose reduction using iterative reconstruction (IR) algorithms remains limited for abdominal CT examinations in the clinical routine as they impact the detection and characterization of low-contrast liver lesions [2,3].

New deep-learning reconstruction (DLR) algorithms have been developed; these can reduce the noise magnitude without altering the image texture, which is very promising for the visualization of low-contrast liver lesions. The first two DLR algorithms featured a deep neural network (DNN) to differentiate the signals from image noise [4,5]. Canon Medical Systems developed a DLR algorithm called AiCE, which features a DNN trained with high-quality model-based IR images from patients. The algorithm developed by GE Healthcare, TrueFidelity^TM^, features a DNN trained with high-quality filtered back-projection (FBP) images from patients and phantoms. Studies performed on the first two DLR algorithms have shown a better detection of these lesions than when using an IR algorithm at the same dose level [6,7,8,9,10,11,12,13,14] or at a lower dose level [15,16,17,18]. Recently, Philips Healthcare also developed an artificial intelligence DLR algorithm called Precise Image [4,19,20]. The algorithm uses a convolutional neural network (CNN), a subtype of the DNN in which each layer performs a convolution operation. The CNN is trained to reproduce the image appearance (noise magnitude and noise texture) of routine-dose FBP images from the raw data of low-dose CT scans. To prevent patients from being overexposed, low-dose images are generated from routine-dose images using a simulation technique to accurately model photon and electronic noise. According to the manufacturer, the CNN was validated by comparing low-dose images generated by AI-DLR with routine-dose images reconstructed using standard methods. Preliminary phantom studies using the Precise Image algorithm have shown that abdomen–pelvic CT examinations are optimized without modifying the image texture [20]. Radiologists have validated the overall quality of the abdominal images for a CT dose index volume (CTDI_vol_) of close to 6 mGy using the Smooth level of this algorithm.

The objective of this study was to assess the image quality and conspicuity of liver metastases of low-dose abdominopelvic CTs performed during routine follow-up for a 6-mGy CTDI_vol_ using this new Precise Image algorithm compared with an IR algorithm.

## 2. Materials and Methods

### 2.1. Patients

From December 2021 to February 2022, all consecutive eligible adult patients with at least one liver metastasis diagnosed from a previous CT scan and who underwent a follow-up CT scan within the inclusion period were enrolled. For these patients, the CT protocol commonly used at our institution was used with the conventional acquisition and reconstruction parameters. However, the raw data were also retrospectively reconstructed with the new DLR algorithm.

This retrospective single-center study was approved by our institutional review board. Participants (and/or their legal guardians) were systematically informed that their data were being collected for an anonymous retrospective study and that they could refuse to participate in the study (i.e., non-opposition statement). No formal calculation of the number of subjects required was performed in accordance with the feasibility and initial experience of the study design.

### 2.2. CT Protocol

Acquisitions of the abdomen–pelvis were made on an Incisive CT Premium scanner (Philips Healthcare). This CT system is equipped with the fourth generation of the hybrid IR algorithm iDose (iDose^4^) and the Precise Image DLR algorithm. The characteristics and overall working principle of this new DLR algorithm have already been defined in several studies [4,19,20,21].

Acquisitions of the abdomen–pelvis were performed at the portal phase 70 s after the beginning of low-osmolar iodinated contrast media injection. The iodinated contrast media were injected with a standard power injector at an injection rate of 3–5 mL/s, and the total volume injected was adjusted according to the body weight (2 mL/kg).

The acquisition parameters were as follows: a tube voltage of 100 kVp (120 kVp for overweight patients), pitch factor of 1, rotation time of 0.35 s/rot, and physical beam collimation of 64 × 0.625 mm. The automatic tube current modulation system was used with a dose right index set at 15 to be close to the recommended 6 mGy CTDI_vol_ [20].

Raw data were reconstructed using level 4 of the iDose^4^ IR algorithm (i4) and the Standard, Smooth, and Smoother levels of the Precise Image (DLR) algorithm. The reconstruction kernel “B” was used for i4, and the “Soft tissue” reconstruction kernel was used for the DLR algorithm. To compare the image quality obtained with the different reconstruction algorithms, a 1 mm slice thickness was used for all images.

### 2.3. Dosimetry Evaluation

The CTDI_vol_, dose length product, size-specific dose estimate, and average scan size calculated by the CT system were retrieved for each patient from the review report at the end of the acquisitions.

### 2.4. Metastatic Evaluation

Metastatic liver disease was classified according to the number and size of metastases (Table 1). The limits were those classically recognized in the literature as influencing curative (thermoablation or surgery) or palliative management [22,23]. The conspicuity of metastasis was determined using the five-point Likert scale defined by Nakamura et al., where 1 = definite artifact mimicking a lesion; 2 = probable artifact mimicking a lesion; 3 = subtle lesion; 4 = well-visualized lesion with poorly visualized margins; and 5 = well-visualized lesion with visualized margins [12]. All images were evaluated by a senior radiologist with 10 years of experience (R1; JF) and who was blinded to the reconstruction type.

### 2.5. Objective Image Quality Assessment

All image quality objective assessments were performed by a junior radiologist with 5 years of experience (R2; QD) on the manufacturer workstation (IntelliSpace Portal, Philips Healthcare, Amsterdam, the Netherlands). Four regions of interest were placed in the muscle, normal liver, largest liver lesion, and fat. The mean *(N_CT_*) and standard deviation (noise) of pixel values were computed, and the contrast-to-noise ratio (*CNR*) was calculated as follows:$$CNR=\frac{\left|N_{CT,tumor}-N_{CT,liver}\right|}{{Noise}_{fat}}$$

### 2.6. Subjective Image Quality Assessment

All abdominal images were read by the two radiologists (R1 and R2) who were blinded to each other’s interpretation and the reconstruction type (algorithm and levels). They assessed the image noise and smoothing using a commonly used five-point scale [19] where 1 = excellent, 2 = above average, 3 = acceptable, 4 = suboptimal, and 5 = unacceptable. The overall image quality was also rated using a previously published scale [19,20] where 1 = not evaluable, 2 = interpretable despite moderate artifacts or noise, 3 = fully interpretable with mild artefacts or noise, and 4 = no artifacts or noise.

### 2.7. Statistical Analyses

Statistical analyses were performed by an in-house biostatistician (C.S) using SAS v9.4 software. For all quantitative data, normality was explored graphically and through the Shapiro–Wilk test [24]. Quantitative data were expressed as means ± standard deviations (SD) and medians with first and third quartiles when appropriate.

Differences in the N_CT_, image noise, *CNR* values, and ordinal variables between i4 and all DLR levels were determined using the two-tailed Wilcoxon signed-rank test. Significance was set up at *p*-value < 0.05. The agreement between the two radiologists was estimated using Gwet’s AC2 with its 95% confidence interval [25,26]. An estimate of <0.4 was considered as a poor agreement, from 0.4 to 0.6 as fair, from 0.6 to 0.8 as good, and >0.8 as an excellent agreement [25]. When both readers rated an identical score for all images, the lack of variance did not allow for a concordance coefficient to be calculated, which is indicated as “-” in Table 2.

## 3. Results

### 3.1. Patients

The study included 30 patients with known hepatic lesions. There were 13 (43%) women and 17 men (57%), and the mean age was 70.4 ± 9.8 years. The mean amount of iodinated contrast material injected was 84.1 ± 11.9 mL.

The primary neoplasm was a colorectal cancer (*n = 18, 60%*), pancreatic cancer (*n = 4, 13%*), breast cancer (*n = 3, 10%*), renal cancer (*n = 3, 10%*), or an ovarian cancer (*n = 2, 7%*) (Table 1).

### 3.2. Dosimetry

For the abdomen–pelvis CT acquisition at the portal phase, the mean CTDI_vol_ was 6.3 ± 2.1 mGy, and the dose length product was 314.7 ± 105.7 mGy.cm (Table 1). The mean scan size was 29.9 ± 3.0 cm, and the size-specific dose estimate was 7.5 ± 1.7 mGy. The mean tube voltage used was 108 ± 10 kVp, and the tube voltage at 120 kVp was used for 12 patients with the highest mean scan size.

### 3.3. Objective Image Quality Assessment

For the four tissues assessed, the mean CT attenuation was similar between i4 and DLR and between all DLR levels (*p* > 0.05) (Table 2 and Table 3).

For all tissues, the image noise was significantly lower with the Standard level of the DLR algorithm than with i4 (−22 ± 10%; *p* < 0.05), but this also was the case for Standard compared with Smooth (−26 ± 10%) and for Smooth compared with Smoother (−37 ± 8%) DLR levels (*p* < 0.05). The opposite pattern was found for the *CNR* (*p* < 0.05).

### 3.4. Subjective Image Quality Assessment

The two radiologists found that the image noise significantly decreased between the i4 and Standard (*p* < 0.001) DLR level and from the Standard to the Smoother level (*p* < 0.05) (Table 2 and Table 3). The opposite pattern was found for image smoothing (*p* < 0.05). Both radiologists rated the images as “No image noise” for all patients with the Smoother level; they rated the image smoothing with the same score for all patients with i4, Standard, and Smoother levels. For other reconstruction types, the agreement between the two radiologists was “good” or “excellent”.

The overall image quality score significantly increased from i4 to the Standard level (*p* < 0.05) and from the Standard to the Smoother level (*p* < 0.05) (Figure 1). Both radiologists rated the overall image quality as “Interpretable despite moderate artefacts or noise” for all patients with i4. For all DLR levels, the overall image quality was rated as “Fully interpretable with mild artefacts or noise” or as “No artefacts or noise”. Agreement between the two radiologists was “excellent” for the Standard and Smooth levels. For i4 and the Smoother level, the two radiologists rated the images with the same score for all patients.

The conspicuity score significantly increased from i4 to the Standard level and from the Standard to the Smoother level (*p* < 0.05) (Figure 2 and Figure 3). The conspicuity score was ≥4 for 37% with i4, 53% for the Standard, 80% for the Smooth, and 93% for the Smoother level.

## 4. Discussion

For the first time, this study has assessed the impact of the new deep learning-based image reconstruction algorithm (Precise Image) on the quality of low-dose abdominal CT images and the detection of liver metastases. We found that this algorithm reduced the image noise and improved the liver’s contrast-to-noise ratio, overall image quality, and liver lesion conspicuity compared with the iDose^4^ iterative reconstruction algorithm.

The image quality obtained with the dose level defined in the preliminary phantom study was considered to be clinically sufficient for the detection and follow-up of liver lesions with the DLR algorithm [20]. For all DLR levels, the overall image quality was rated as fully interpretable with or without mild artefacts or noise with a good or excellent agreement between the two radiologists. This low-dose level was close to 6 mGy and was lower than in most of the studies with the two other DLR algorithms [6,7,8,9,10,11,12,13,16,17]. For these studies, standard dose levels ranged from 10.3 to 28.3 mGy [6,7,8,9,10,11,12,13] and low-dose levels ranged from 8.6 to 11.3 mGy. However, the dose level used was higher than the ultra-low dose level used by Noda et al. [17] for pancreatic CT or by Singh et al. [18] and Cao et al. [16] for contrast-enhanced abdominal CT.

The study outcomes confirmed the results found in the phantom preliminary study [20]. The objective and subjective assessments of the image noise showed that it decreased as the DLR levels increased (from Standard to Smoother) and were lower for all DLR levels than with the IR algorithm for the same HU values of different tissues. Similar results were found in different studies using the other two DLR algorithms as compared with IR algorithms [7,8,10,11,12,14,16,17,27,28]. In these studies, the decrease in image noise combined with unchanged mean attenuation values led to a better *CNR* for the liver using DLR algorithms. The same results were found in this study for the liver *CNR*, which increased from the Standard to the Smoother level. In addition, the radiologists’ subjective image smoothing assessment confirmed that it increased as the DLR levels increased and was higher with all DLR levels than with the IR algorithm. As far as we know, studies performed with the two other DLR algorithms do not assess image smoothing but instead assess image sharpness or blurring [9,16,17,29].

The reduction in image noise as the DLR levels increased led to an increase in both radiologists’ overall image quality scores despite the increase in image smoothing. The overall image quality was higher with all DLR levels than with IR; it was similar to most studies when comparing the two other DLR algorithms with IR algorithms [6,7,8,11,13,16,17,18]. Lesion conspicuity was also higher with the DLR levels than with IR and was better with the Smooth and Smoother levels, as found in most studies on vessels or liver lesions conspicuity [7,8,9,12,17]; the exception is Kaga et al. who reported a decrease in liver lesion conspicuity as the DLR level increased [10].

This study had certain limitations. First, it included a limited number of patients from a single institution using one CT system. Second, the patients included had known liver metastasis during their follow-up. Different outcomes may be found for other liver or abdominal lesions. In addition, this study only focused on the assessment of subjective image quality and the accuracy of diagnosis using size measurements, whereas the number of metastases was not evaluated. A prospective study with a larger patient population is required to confirm the potential of this DLR algorithm. Thus, with significantly reduced noise and a sufficient, suitable image texture, DLR algorithms open the way to new perspectives of significant dose reduction for these patients. Further patient studies are now required to confirm the generalizability of our study results and assess the potential of this new DLR algorithm for detecting abdominal lesions via ultra-low dose CT acquisitions.

## 5. Conclusions

This study confirms that the Precise Image algorithm reduces the image noise and improves the contrast-to-noise ratio, overall image quality, and lesion conspicuity compared with iterative reconstruction algorithms. The highest score of image quality and lowest image noise results were found with the highest level of Precise Image. Low-dose CT acquisitions with Precise Image may now be used in clinical practice for the detection and follow-up of liver metastases.

## Figures and Tables

**Figure 1 diagnostics-13-01182-f001:**
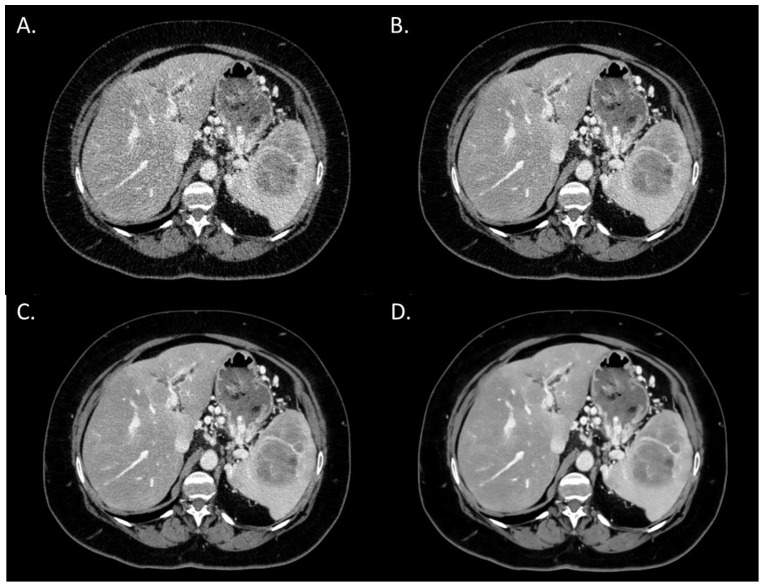
Overall image quality of abdominal CT images (WL: 60 HU; WW: 360 HU) of a woman with pancreatic cancer and abdominal metastatic lesions (63 years old; SSDE: 9.09 mGy; average scan size 32.2 cm). (**A**) iDose^4^ level 4; average overall image quality score: 2.5; (**B**) Precise Image, Standard; average overall image quality score: 3; (**C**) Precise Image, Smooth; average overall image quality score: 4; (**D**) Precise Image, Smoother; average overall image quality score: 4.

**Figure 2 diagnostics-13-01182-f002:**
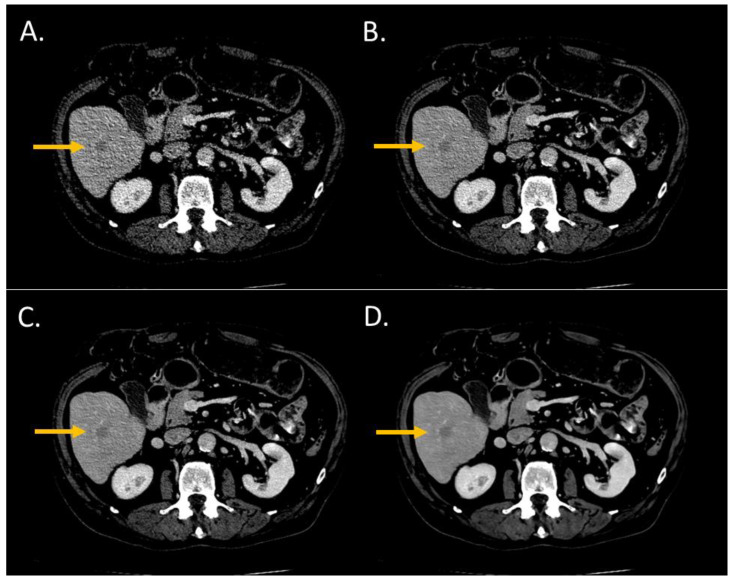
Lesion conspicuity score of liver CT images (WL: 90 HU; WW: 190 HU) of a man (64 years old, SSDE: 6.73 mGy; average scan size 28.9 cm) with colorectal cancer and a liver metastasis of 14.1-mm diameter in the segment VI. (**A**) iDose^4^ level 4; lesion conspicuity score: 3; (**B**) Precise Image, Standard; lesion conspicuity score: 3; (**C**) Precise Image, Smooth; lesion conspicuity score: 4; (**D**) Precise Image, Smoother; lesion conspicuity score: 5.

**Figure 3 diagnostics-13-01182-f003:**
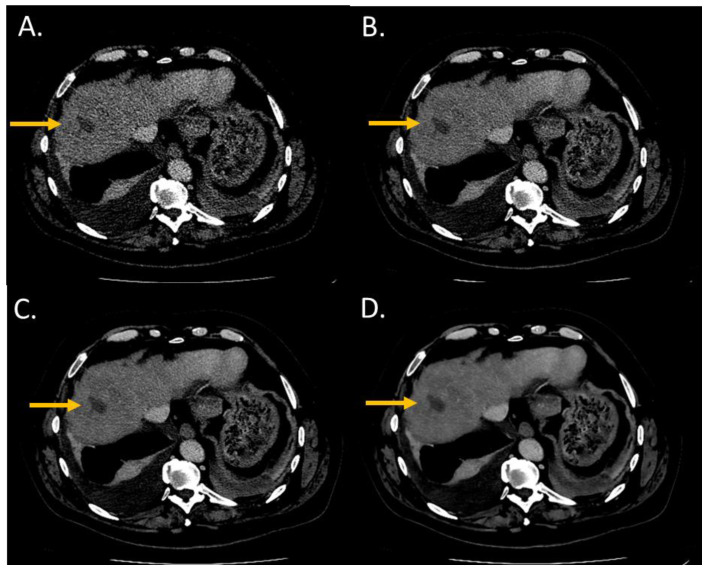
Lesion conspicuity score of liver CT images (WL: 90 HU; WW: 190 HU) of a man (64 years old, SSDE: 10.24 mGy; average scan size 34.6 cm) with a pancreatic cancer and liver metastasis of 2.70 mm of large axis in segment VIII. (**A**) iDose^4^ level 4; lesion conspicuity score: 3; (**B**) Precise Image, Standard; lesion conspicuity score: 3; (**C**) Precise Image, Smooth; lesion conspicuity score: 4; (**D**) Precise Image, Smoother; lesion conspicuity score: 5.

**Table 1 diagnostics-13-01182-t001:** Patients’ characteristics and dosimetry data.

		Values
Age (Years)		70.4 ± 9.8 [40–87]
Sex (Women/Men)		13 (43%)/17 (57%)
Patients on chemotherapy treatment		25 (83%)
Hepatic metastases from	Colorectal cancer	18 (60%)
	Pancreatic cancer	4 (13%)
	Breast cancer	3 (10%)
	Renal cancer	3 (10%)
	Ovarian cancer	2 (7%)
Number of liver metastases	1 to 3	12 (40%)
	3 to 10	9 (30%)
	>10	9 (30%)
Size of liver metastases	<1 cm	1 (3%)
	1 to 3 cm	16 (53%)
	3 to 10 cm	12 (40%)
	>10 cm	1 (3%)
Average size diameter (cm)		29.9 ± 3.0 [23.7–35.9]
Size-specific dose estimate (mGy)		7.5 ± 1.7 [4.8–12.0]
Volume CT dose index (mGy)		6.3 ± 2.1 [3.12–12.27]
Dose length product (mGy.cm)		314.7 ± 105.7 [156.0–613.5]
Amount of iodine injected (mL)		84.1 ± 11.9 [60–110]

Values are expressed as means ± standard deviations [min–max] or the number of patients (percentage).

**Table 2 diagnostics-13-01182-t002:** Objective and subjective image quality assessment.

			iDose^4^ Level 4	Standard	Smooth	Smoother
Objective image quality assessment	Mean attenuation (HU)	Fat	−103.4 ± 12.5	−103.6 ± 12.4	−104.3 ± 11.9	−103.5 ± 11.7
		Muscle	50.0 ± 12.6	49.6 ± 13.6	50.5 ± 12.9	50.1 ± 13.2
		Liver	97.5 ± 16.4	96.8 ± 16.2	97.9 ± 17.4	97.4 ± 17.3
		Tumor	52.3 ± 20.0	51.1 ± 20.4	52.1 ± 21.0	51.5 ± 21.5
	Image noise (HU)	Fat	23.9 ± 5.4	19.5 ± 5.0	14.1 ± 4.0	8.9 ± 2.6
		Muscle	23.8 ± 4.2	19.1± 4.7	14.7 ± 4.8	9.5 ± 6.2
		Liver	24.6 ± 4.3	18.3 ± 3.7	12.9 ± 2.5	7.1 ± 1.5
		Tumor	27.8 ± 6.0	21.3 ± 4.8	15.6 ± 4.0	10.6 ± 5.0
	*CNR*	Liver	2.0 ± 1.0	2.5 ± 1.4	3.5 ± 1.8	5.5 ± 2.7
Subjective image quality assessment	Image noise	Score	4.0 [4.0; 4.0]	3.0 [2.5; 3.0]	2.0 [1.5; 2.0]	1.0 [1.0; 1.0]
		Gwet AC2 [95% CI]	0.86 [0.75; 0.97]	0.65 [0.39; 0.92]	0.78 [0.64; 0.92]	-
	Image smoothing	Score	1.0 [1.0; 1.0]	3.0 [3.0; 3.0]	4.0 [4.0; 5.0]	5.0 [5.0; 5.0]
		Gwet AC2 [95% CI]	-	-	0.71 [0.46; 0.95]	-
	Overall image quality	Score	2.0 [2.0; 2.0]	3.0 [3.0; 3.0]	4.0 [3.0; 4.0]	4.0 [4.0; 4.0]
		Gwet AC2 [95% CI]	-	0.97 [0.90; 1.00]	0.81 [0.60; 1.00]	-
Conspicuity		Average score	3.0 [3.0; 4.0]	4.0 [3.0; 4.0]	4.0 [4.0; 4.0]	4.0 [4.0; 5.0]

Quantitative values are expressed as means ± standard deviations. Qualitative values are expressed as medians [1st quartile; 3rd quartile]. For the Gwet AC test, the 95% CI corresponds to the confidence interval at 95%.

**Table 3 diagnostics-13-01182-t003:** *p*-values calculated for all variables between level 4 of the iDose^4^ algorithm and all Precise Image levels.

Comparison	i4 vs. Standard	i4 vs. Smooth	i4 vs. Smoother	Standard vs. Smooth	Standard vs. Smoother	Smooth vs. Smoother
N_CT_ Liver	0.20	0.91	0.70	0.47	0.08	0.45
N_CT_ Liver tumor	0.12	0.35	0.87	0.47	0.12	0.16
N_CT_ Muscle	0.77	0.86	0.56	0.92	0.71	0.76
N_CT_ Fat	0.66	0.47	0.96	0.87	0.36	0.26
Noise Liver	<0.0001	<0.0001	<0.0001	<0.0001	<0.0001	<0.0001
Noise Liver tumor	<0.0001	<0.0001	<0.0001	<0.0001	<0.0001	<0.0001
Noise Muscle	<0.0001	<0.0001	<0.0001	<0.0001	<0.0001	<0.0001
Noise Fat	<0.0001	<0.0001	<0.0001	<0.0001	<0.0001	<0.0001
Image noise	<0.0001	<0.0001	<0.0001	<0.0001	<0.0001	<0.0001
Image smoothing	<0.0001	<0.0001	<0.0001	<0.0001	<0.0001	<0.0001
Objective image quality	<0.0001	<0.0001	<0.0001	<0.0001	<0.0001	<0.0001
Conspicuity	<0.01	<0.0001	<0.0001	<0.0001	<0.0001	<0.01

N_CT_ corresponds to the mean attenuation. *p*-values lower than 0.05 were considered significant.

## Data Availability

The data presented in this study are available from the corresponding author upon reasonable request.

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
