# Peer review of "First Results of a New Deep Learning Reconstruction Algorithm on Image Quality and Liver Metastasis Conspicuity for Abdominal Low-Dose CT"

_diagnostics, 2023, doi:10.3390/diagnostics13061182_

Round 1

Reviewer 1 Report

In this study, the authors wished to assess the impact of a deep-learning image reconstruction algorithm (Precise Image; 12 DLR) on image-quality and liver metastasis conspicuity compared to an iterative reconstruction algorithm (IR).

The paper is well sounded and the method is neat. The results are clearly reported.

I have the few minor comments.

Single-center is more correct than monocentric. I suggest making the change

In general, the city and country of vendors should be added

The authors should report how the normality of distribution of quantitative variables was made and report variables as medians or means as appropriate. I suggest to add the following reference

Barat M, et al. doi: 10.1016/j.diii.2022.09.007.

The authors should indicate if the tests were two-tailed or not.

In the Discussion section, the authors should add some other applications of deep learning reconstruction to obtain greater image quality with CT and give more importance to this algorithm. I suggest the folllowing references

doi: 10.1016/j.diii.2021.08.001. and doi: 10.1016/j.diii.2021.03.002.

As a limitation, but this is understandable, the authors should acknowledge that the analysis was restricted to image quality.

Refs 18 should be updated

Figure 3 is a little too dark

Author Response

In accordance with the reviewer's comment, our manuscript has been proofread by our medical writer (native English speaker).

Comments and Suggestions for Authors

In this study, the authors wished to assess the impact of a deep-learning image reconstruction algorithm (Precise Image; 12 DLR) on image-quality and liver metastasis conspicuity compared to an iterative reconstruction algorithm (IR).

The paper is well sounded and the method is neat. The results are clearly reported.

We thank the reviewer for these positive comments.

I have the few minor comments.

Single-center is more correct than monocentric. I suggest making the change

This correction has now been made.

In general, the city and country of vendors should be added

We thank the reviewer for pointing this out. However, after checking several of our articles published in MDPI, there is no obligation to add the city and country of vendors.

The authors should report how the normality of distribution of quantitative variables was made and report variables as medians or means as appropriate. I suggest to add the following reference

Barat M, et al. doi: 10.1016/j.diii.2022.09.007.

The authors should indicate if the tests were two-tailed or not.

We thank the reviewer for this comment. In the "Statistical analysis" section, we have specified the methods for assessing normality as follows: "For all quantitative data, normality was explored graphically and through the Shapiro-Wilk test [24]. Quantitative data were expressed as means ± standard deviations (SD) and medians with first and third quartiles when appropriate."

Following the use of this test, we have also replaced means +/- SD with medians and quartiles for qualitative values in Table 2.

As for the tests, these were two-tailed, as indicated "using the two-sided Wilcoxon signed-rank test". To improve clarity, we have now replaced the word "sided" by "tailed".

In the Discussion section, the authors should add some other applications of deep learning reconstruction to obtain greater image quality with CT and give more importance to this algorithm. I suggest the folllowing references

doi: 10.1016/j.diii.2021.08.001. and doi: 10.1016/j.diii.2021.03.002.

 We thank the reviewer for this most helpful comment. We have added the first reference in the introduction and also in the Discussion section. However, we cannot add the 2nd reference because it deals with the Siemens iterative reconstruction algorithm (ADMIRE) and not with a DLR algorithm.

As a limitation, but this is understandable, the authors should acknowledge that the analysis was restricted to image quality.

In accordance with the reviewer’s comment, we have now added the following sentence in the limitations part: “In addition, this study only focused on the assessment of subjective image quality and the accuracy of diagnosis using size measurements whereas the number of metastases was not evaluated”.

Refs 18 should be updated

We thank the reviewer for pointing this out and have now updated the reference.

Figure 3 is a little too dark

We understand the reviewer's comment. However, this figure was made using the windowing typically used in the clinic to visualize CT images of the liver (WL: 90 HU; WW: 190 HU). Therefore, we feel that it is more appropriate to leave the images in this figure with this windowing as it is more representative.

Reviewer 2 Report

Authors have not either explained or demonstrated the details of Deep Learning network effectively. I advise the authors to rewrite the paper based on DL network employed for reconstruction, give precise architecture, and explain the analysis effectively. 

Author Response

In accordance with the reviewer's comment, our manuscript has now been proofread by our medical writer (native English speaker).

Comments and Suggestions for Authors

Authors have not either explained or demonstrated the details of Deep Learning network effectively. I advise the authors to rewrite the paper based on DL network employed for reconstruction, give precise architecture, and explain the analysis effectively.

We thank the reviewer for this comment and have added explanations about the operation of all DLR algorithms used in this study and, in particular, the Precise Image algorithm.

Round 2

Reviewer 2 Report

Authors have not provided a point by point response for the comments given in the last round. They have uploaded the manuscript document as response letter. Hence i could  not review further.